# Load Balancing Based on Firefly and Ant Colony Optimization Algorithms for Parallel Computing

**DOI:** 10.3390/biomimetics7040168

**Published:** 2022-10-17

**Authors:** Yong Li, Jinxing Li, Yu Sun, Haisheng Li

**Affiliations:** 1School of E-Commerce and Logistics, Beijing Technology and Business University, Beijing 100048, China; 2School of Computer Science and Engineering, Beijing Technology and Business University, Beijing 100048, China

**Keywords:** firefly algorithm, bio-inspired design, hybrid methods, load balance, heuristic algorithm, multi-objective optimisation, parallel computing

## Abstract

With the wide application of computational fluid dynamics in various fields and the continuous growth of the complexity of the problem and the scale of the computational grid, large-scale parallel computing came into being and became an indispensable means to solve this problem. In the numerical simulation of multi-block grids, the mapping strategy from grid block to processor is an important factor affecting the efficiency of load balancing and communication overhead. The multi-level graph partitioning algorithm is an important algorithm that introduces graph network dynamic programming to solve the load-balancing problem. This paper proposed a firefly-ant compound optimization (FaCO) algorithm for the weighted fusion of two optimization rules of the firefly and ant colony algorithm. For the graph, results after multi-level graph partitioning are transformed into a traveling salesman problem (TSP). This algorithm is used to optimize the load distribution of the solution, and finally, the rough graph segmentation is projected to obtain the most original segmentation optimization results. Although firefly algorithm (FA) and ant colony optimization (ACO), as swarm intelligence algorithms, are widely used to solve TSP problems, for the problems for which swarm intelligence algorithms easily fall into local optimization and low search accuracy, the improvement of the FaCO algorithm adjusts the weight of iterative location selection and updates the location. Experimental results on publicly available datasets such as the Oliver30 dataset and the eil51 dataset demonstrated the effectiveness of the FaCO algorithm. It is also significantly better than the commonly used firefly algorithm and other algorithms in terms of the search results and efficiency and achieves better results in optimizing the load-balancing problem of parallel computing.

## 1. Introduction

Computational fluid dynamics (CFD) [1] is a discipline that solves and analyzes fluid dynamics problems through computer and numerical simulation methods. In the process of CFD numerical simulations, the complexity of the problem is gradually increasing, the accuracy of the numerical simulation is becoming more and more demanding, and the network size is continuously expanding. In the process of numerical simulations, the computational area is discretized into multiple grid blocks of different sizes, and these blocks are assigned to different processors for parallel computation. Therefore, how to make the load task of each processor reasonably balanced is the main problem to be solved, and it is also an important technology to improve the processing efficiency. As the scale of computation increases, the impact of the communication overhead of the processors is also gradually increasing on the parallel processing efficiency. The mismatch between the number of grid blocks and the number of processes, as well as the mismatch between the computational capacity of grid blocks and the computational capacity of processes, makes the traditional partitioning or combination strategies unable to meet the demand of load balancing well [2]. Therefore, the research on algorithms for large-scale load balancing is crucial.

At present, load-balancing algorithms for CFD parallel computing are being gradually and extensively researched in the world. For example, the load-balancing algorithm of greedy method and the recursive pairwise edge splitting load-balancing algorithm proposed by Streng [3] and Ytterström [4] are classical load-balancing algorithms based on a geometric level. In addition, Hendrickson et al. [5] proposed a multi-level algorithm for partitioning graphs, which is a method of mapping to a refined graph by coarsening the graph dissection. In recent years, heuristic algorithms have also been increasingly used in the research of load-balancing algorithms. Most of them are swarm intelligence algorithms that imitate natural bodies, such as firefly algorithm, ant colony algorithm, and bee colony algorithm. They are applied to solve various optimization scheduling problems and are widely used in industry [6], network transmission [7], biology [8] and cloud computing [9]. Among them, the combination of neural networks and other algorithms has achieved considerable research success in image capture and retrieval [10,11,12,13]. With the development of parallel computing, the improvement and optimization of heuristic algorithms to solve the load-balancing problem of parallel computing have also achieved considerable results. For example, Kernighan et al. [14] proposed a heuristic algorithm for graph segmentation. The hybrid load-balancing algorithm proposed by Yang Chengfu et al. [15] and the ant colony optimization based MrLBA algorithm proposed by Arfa Muteeh [16] are typical methods of heuristic algorithms applied to load balancing. However, swarm intelligence algorithms generally easily fall into local optimum, and the optimization effect is uneven. This paper uses the multi-level graph segmentation algorithm of a structural grid to obtain a network coarsening graph and proposes a FaCO algorithm based on the firefly and ant colony algorithm to subdivide the coarsening graph. Based on the fusion of the two in the optimization rule, a new optimization rule is established to adjust the position update, which avoids falling into the local optimal situation to a certain extent.

In this paper, we take TSP public datasets such as Oliver30 and eil51 to conduct experiments, and the results demonstrate that the FaCO algorithm is significantly better than the commonly used firefly algorithm in terms of finding the optimal results. The number of iterations is reduced, and the efficiency of finding the optimal results is higher with the same number of iterations. In addition, the optimal solution is obtained with the same number of iterations, and the algorithm does not fall into local optimum, which solves the problem that the firefly algorithm frequently falls into local optimum to a certain extent and thus optimizes the load-balancing problem in parallel computing.

The rest of this paper is organized as follows. Section 2 presents the related work, followed by Section 3, which presents the background and problem statement. Then, Section 4 presents the methods and models, the proposed method is described in Section 5, experiments and analysis are presented in Section 6, and finally conclusions and future work are drawn in Section 7 and Section 8, respectively.

## 2. Related Work

In order to solve the time and cost problems of large-scale computation, parallel computing decomposes complex problems into several parts with certain regularity and assigns each part to a separate processor for simultaneous computation of multiple instructions. It not only reduces the time cost but also improves the overall computational performance. With the development of CFD numerical simulation research, the scale of computation is increasing, and parallel computing becomes an effective method to solve this problem.

With the wide application of CFD numerical simulations, such as point cloud sampling [17], multi-view image retrieval [18], etc., the research on load-balancing algorithms for CFD parallel computing can be basically divided into two main categories: geometry-based and graph-based [3,4,5]. However, with the development of research on heuristic algorithms, there is a great potential for their application to solve load-balancing problems. We mainly focus on the improvement of heuristic algorithms and their graph-based optimization applications to load-balancing problems for parallel computing.

Kannan et al. proposed a multi-objective load-balancing method using a bio-inspired algorithm [19]. It solves the pre-convergence problem by a micro-genetic algorithm and proposed a stable method of combining cat swarm optimization for process load distribution (MG-CSO). Better results are obtained on the time cost problem of load balancing for cloud computing.

The hybrid discrete artificial bee colony algorithm (ABC) proposed by Junqing Li and Yunqi Han investigated and solved the task scheduling problem in cloud computing systems [20]. It designs an improved scout bee using different local search methods to obtain the best food source or waste solution that can improve the convergence of the proposed algorithm.

Ahmad M. Manasrah et al. proposed a hybrid algorithm based on genetic algorithm (GA) and particle swarm optimization (PSO) to solve the multitask scheduling problem [21]. In the field of multitask scheduling, this algorithm converges to the optimal solution much faster and with higher quality.

The firefly load-balancing algorithm was proposed by Manisha et al. It reduces the computational cycles and the degree of load imbalance while exhibiting better working performance [22]. Both the genetic ant colony algorithm proposed by Cheng Cheng et al. [23] and the ACO focusing algorithm proposed by Skinderowicz Rafał [24] improved the ant colony algorithm to a new level and obtained better computational performance. Tang Bo et al. proposed the idea of applying genetic algorithms to the mapping process of grid blocks and processors and then performing intelligent allocation [2], but genetic algorithms have more space for optimization than other algorithms.

In the optimization process of a series of heuristic algorithms, although they also solve the load-balancing problem to a certain extent, they have the limitation of a single mechanism of finding the best and the tendency to fall into the local optimum. In this paper, the proposed FaCO algorithm based on the fusion of firefly algorithm and ant colony algorithm with merit-seeking mechanism is free from the constraints of a single mechanism. Additionally, the previous heuristic algorithm is introduced to fuse the allocation mapping process of parallel computing, and the innovative FaCO algorithm is applied to solve the simple tsp graph structure after multi-level graph dissection. The load-balancing problem of massively parallel computing is solved with the goal of optimal allocation.

## 3. Background and Problem Statement

### 3.1. Background

#### 3.1.1. Parallel Computing

Parallel computing is proposed in contrast to traditional serial computing, which works simultaneously like a parallel circuit. When a problem (e.g., numerical simulation) is decomposed into a series of discrete parts that can be executed concurrently, each part can also be decomposed into a series of discrete instructions, and the discrete instructions of each part can be executed simultaneously on different processors. This can greatly reduce computation time and increase computational efficiency.

Parallel computing is a subdivision of high-performance computing, which decomposes complex problems into several parts with a certain pattern and assigns each part to a separate processor for simultaneous computation of multiple instructions. It is used to solve large-scale computational problems and to improve the speed and performance optimization of computation. The basic evaluation metrics for parallel computing are execution time, workload, and communication overhead. With the development of CFD numerical simulation research, its computational scale is increasing, and parallel computing becomes an effective method to solve this problem. However, there are certain problems in large-scale parallel computing: each processor has a certain load limit, and if the allocated computational units exceed or are far less than the load of the processor, then it will cause the problem of unprocessable or wasted resources. How to allocate computation units to processors in a reasonable and efficient way and minimize the communication overhead between processors is the main content of the load-balancing problem.

#### 3.1.2. Gragh and Bioinspired Algorithms

A graph consists of an infinite nonempty set of vertices and a set of edges between the vertices, usually denoted as: G = (V, E). V is the set of vertices and E is the set of edges. When the edges between two vertices have no direction, then it is called an undirected graph. We mainly study simple undirected graphs in this paper.

Inspired by various natural phenomena and various behaviors of biological populations, humans have proposed new methods to solve many complex optimization problems, such as the firefly algorithm, which simulates the light-attracting motion of fireflies, and the ant colony algorithm, which finds the shortest path from the food source to the ant nest when foraging. These optimization methods inspired by biological behavior are collectively known as bio-inspired algorithms. Bionic computing is the computer modeling of this biological systems to solve various problems using the characteristics of biological populations. The TSP problem is a typical application area of bionic algorithms.

### 3.2. Problem Statement

Therefore, we will proceed with the following problem statement based on the background of Section 3.1.

During the numerical simulation, we consider the processors as the vertices of the graph and the edge weights of each vertex as the communication overhead between processors. Since there is no static order and direction of communication between processors, the parallel computation of processors is abstracted into an undirected graph structure. The parallel computation of processors is abstracted into a graph structure, but the graph is still very complex and large. Using a multilevel graph dissection algorithm to coarsen the complex original graph into a structurally simple graph [25], to achieve load balancing of the graph, it is necessary to minimize the sum of edge weights through each vertex, which is the TSP problem.

For the solution of the TSP problem of the coarsened simple graph, we innovatively proposed an optimal combination of the optimization mechanism by the firefly and ant colony algorithm to map the optimized result graph back to the original graph, that is, to solve the load-balancing problem of parallel computing. Although there have been many improvements and optimizations of heuristic algorithms before this, they are all based on the improvement of a single merit-seeking mechanism and have certain limitations. The FaCO algorithm proposed in this paper not only breaks away from the limitations of the single search mechanism but also provides significant improvements in the search results and efficiency. We also innovatively proposed to combine the improved algorithm with the multilevel graph dissection algorithm, which simplifies and abstracts the complex parallel computation problem into a simple graph, maintaining the global search capability of the heuristic algorithm and also reducing the spatial search breadth. Additionally, the mapping strategy of the graph dissection algorithm ensures the influence of the resulting search results on the initial complex problem.

## 4. Methods and Models

In this process, the related multi-level graph dissection algorithm, firefly algorithm and ant colony algorithm are highlighted in this paper.

### 4.1. Multi-Level Algorithm for Partitioning Graphs

The multi-level algorithm for partitioning graphs is divided into three steps, coarsening, dissecting the coarse graph, and coarse graph projection. The coarsening process is to divide the original graph $G_{0}=\left(V_{0},E_{0}\right)$ into subgraphs, with each subgraph as a vertex of the coarsened graph and according to a certain mapping strategy: (1)$$f_{i}=V_{i}\to V_{i+1}$$

Roughening occurs here according to the compression ratio: (2)$$\frac{V_{i}}{V_{i+1}}=\rho ,\rho >1$$

The coarsening process is shown in Figure 1.

The overhead of both the multilevel graph dissection construction coarsening graph and the local refinement algorithm is proportional to the number of edges in the graph. In the coarsest granularity graphs, moving nodes in different profiles correspond to moving a series of nodes simultaneously in the original graph. Therefore, when traversing this series of graphs at different granularities, better improvements can be continuously made to the final profile [26].

### 4.2. Firefly Algorithm

The firefly algorithm is a bionic firefly population search algorithm, which takes advantage of the feature that the strong fireflies attract the weak fireflies [27]. In the process of moving the weak fireflies to the strong fireflies, the iteration of positions is completed so as to find the optimal position, that is, to complete the process of finding the optimum. After coarsening to get the coarsest graph, the need to ensure that the shortest distance between the connection of points (communication overhead is minimal) and adjacent path points can be assigned to the same processor. Solving this NP problem is the advantage of the firefly algorithm. The flow chart is shown in Figure 2.

Due to the special nature of the firefly algorithm’s ability to solve the TSP problem, we consider the spatial location of each firefly as a set of solutions. Initially, we randomize m firefly locations and generate the two-dimensional locations of n points according to their edge weights G = (V, E) after obtaining the graph dissection results, which are converted into the TSP problem. Suppose the vertex number is $No=\left\{1,2,\ldots ,n\right\}$; then each firefly represents an arbitrary arrangement of n vertices, i.e., $Route=\left\{r_{1},r_{2},\ldots ,r_{n}\right\}$, and $r_{i}$ represents the i-th city of the route.

Suppose $R=\left\{r_{1},r_{2},\ldots ,r_{n}\right\}$ is a set of solutions of the firefly, so the length of the path is: (3)$$R_{i}=\sum_{i=1}^{n-1}D_{r_{i}r_{i+1}}+D_{r_{n}r_{1}}$$

Let the objective function be: (4)$$f\left(i\right)=R_{i}$$

The smaller the value of the objective function is, the shorter the path length is. The absolute brightness is directly determined by the value of the objective function, which represents the degree of superiority of the solution; then, the absolute brightness formula can be set as: (5)$$I\left(i\right)=\frac{1}{f{\left(i\right)}^{2}}$$

That is, the shorter the distance length, the greater the absolute brightness value of the firefly.
(6)$$r_{ij}={\left(r_{j1}-r_{i1}\right)}^{2}+{\left(r_{j2}-r_{i2}\right)}^{2}+{\left(r_{j3}-r_{i3}\right)}^{2}+\cdots +{\left(r_{jn}-r_{in}\right)}^{2}$$

When the position update requirement is reached, the next moving firefly position value is assigned to the current firefly.

Although the firefly algorithm is widely used in the TSP problem, there are still problems, such as the ease of falling into the local optimum, which leads to a poor search effect. Therefore, we introduce the rules of ant colony algorithm on the basis of firefly algorithm, so that the rules of the two are weighted together, which integrates the advantages of the two search rules and makes them more reasonable and can avoid falling into local optimum to some extent.

### 4.3. Ant Colony Optimization

The basic idea of the ant colony optimization for solving optimization problems is that the paths of ants represent the feasible solutions of the problem to be optimized, and all the paths of the whole ant colony constitute the solution space of the problem to be optimized [28]. The ants with shorter paths release more pheromones, and as time progresses, the concentration of pheromones accumulated on the shorter paths gradually increases, and the number of ants choosing that path also increases. Eventually, all ants will concentrate on the best path under the effect of positive feedback, which corresponds to the optimal solution of the problem to be optimized. The flow chart of the algorithm is shown in Figure 3.

The probabilistic search formula for the ant colony is delineated below.

The probability of ant k visiting node j from node at moment t can be written as follows: (7)$$P_{ij}^{k}=\left\{\begin{array}{cc} \frac{\tau_{ij}^{\alpha }\left(t\right)*\left(1/r_{ij}\right)}{\sum_{s\in unvisit_{k}}\tau_{ij}^{\alpha }\left(t\right)*\left(1/r_{ij}\right)} & j\in \;\mathrm{unvisit}{\;}_{k}\\ 0 & \;\mathrm{other}\; \end{array}\right.$$
where $r_{ij}$ is the distance between i and j at time t, and $\tau_{ij}^{\alpha }\left(t\right)$ is the pheromone concentration from i to j at time t. $unvisit_{k}$ is the set of nodes that have not been visited yet.

The positive feedback mechanism of the ant colony algorithm makes the search process converge continuously and eventually approach the optimal solution. Each individual can change the surrounding environment by releasing pheromones, and each individual can sense the real-time changes of the surrounding environment [29]. The individuals communicate with each other indirectly through the environment. The probabilistic search method of ant colony algorithm does not easily fall into local optimum and instead experiences ease in finding the global optimal solution. It also has other advantages, which makes it an effective heuristic algorithm to solve CFD problems. Therefore, the integration of the pheromone probabilistic iteration rules of the ACO with the firefly algorithm is proved to be a better improvement than the firefly algorithm alone to solve the load-balancing algorithm for massively parallel computation.

## 5. Proposed Method

### 5.1. The FaCO Principle

Although the firefly algorithm and ant colony algorithm are applied to solve the NP problem with significant results, both still have certain defects [30]. Due to the need for too many parameters, which must be set in advance, and the ease of falling into the local optimum, the wide search area and high-dimensional optimization problems have a very weak attractiveness and have difficulty influencing the phenomenon of location updates. Therefore, this paper will contain two algorithms in the location update given the corresponding weight according to the weight calculated after the results of the location update, so that the integrated update results from the two algorithms for optimization. Thus the update efficiency becomes faster, but this also avoids the single algorithm update method leading to local optimum. In this paper, the weights are adjusted according to a BP neural network [31], and the final result error is made smaller by the gradient descent method. Then, the most suitable weights are obtained according to the results.

The main principles are as follows.

When the graph dissection result G = (V, E) forms the initial n location nodes, we initialize the population location according to the firefly algorithm and initialize the firefly related parameters: firefly size m, firefly location graph G′ = (V′, E′), absolute brightness $I_{0}$, maximum attractiveness $\beta_{0}$, and light absorption coefficient γ.

At the same time, the number of firefly populations and their initial positions are used as the number and positions of the new TSP cities, i.e., for the ant colony, the firefly populations form m initial position nodes for the newly generated G′ = (V′, E′) and initialize each parameter of the ant colony: colony size p, pheromone constant Q, pheromone factor α, heuristic function factor β′, and pheromone volatility factor ρ.

For the i-th firefly, i.e., the i-th city of the colony, we calculate the probability of each point according to the current pheromone of the colony using roulette wheel selection (see Equation (Equation 7)) and select the point with the highest probability as the alternative point $k_{i}$ for the next city to be selected in the current situation while judging the highest brightness point within the decision radius in the firefly algorithm as the alternative point $j_{i}$ for the next location to be migrated.

The relative brightness of fireflies can be calculated as follows: (8)$$I=I_{0}e^{-\gamma r}$$

We give $k_{i}$, $j_{i}$ weights as $W_{1}$ and $W_{2}$, respectively, and obtain the weighted index value $g_{i}$: (9)$$g_{i}=W_{1}*k_{i}+W_{2}*j_{i}$$

The index of the closest point $u_{i}$ to $g_{i}$ in the remaining firefly index is selected as the next location migration point for the i-th firefly.
(10)$$u_{i}=\min\left\{r_{iu}=\left|X_{u}-X_{q}\right|\right\}q=1,2,\ldots \ldots ,i-1,i+1,\ldots \ldots ,m$$

The position of the i-th element at moment t is $X_{i}\left(t\right)$; then, the actual updated position of the firefly at moment t + 1 after anthroposophic weighting of the i-th element is:(11)$$X_{i}\left(t+1\right)=X_{u}\left(t\right)$$

The initial parameters of the FaCO algorithm are set as in Table 1.

### 5.2. Algorithm Flow

In this section we focus on the algorithmic logic and flow of the FaCO algorithm.

#### 5.2.1. Algorithm Logic Diagram

The logic diagram of the FaCO algorithm is shown in Figure 4. It mainly shows the logical rules of FaCO algorithm, which is the logical basis for the fusion of Firefly and Ant Colony algorithm mechanisms.

#### 5.2.2. Algorithm Pseudocode

The pseudocode of the FaCO algorithm is shown in Algorithm 1. Firstly all parameters are input for initialization definition and the objective function optimal value is output after a conditional loop.
**Algorithm 1** FaCO algorithm.**Input:** m, I0, β0, γ, p, Q, α, β′, ρ, Max           ▹ Initialization parameters**Output:** global extreme value points and optimal individual values
1:n = 1                           ▹ Number of iterations2:**while** n < Max **do**      ▹ Executes within the maximum number of iterations3:  Calculate the relative brightness I of fireflies and initialize the ant colony pheromone matrix4:  **for**
i=1,2,…,m
**do**5:   **for**
j=1,2,…,m
**do**6:    Calculate the probability of selecting the remaining points when the ant colony departs from point i. Select the departure alternative with the highest probability $k_{i}$, and update the next city selection for ant i as $k_{i}$7:    **if**
$I_{i}<I_{j}$
**then**8:     Calculate the distance nearest point u after weighting $k_{i}$ and j.9:     Update the position of firefly i to u and recalculate the firefly brightness. Update the ant colony pheromone matrix and path parameters.10:    **end if**11:   **end for**12:  **end for**13:  n = n + 114:**end while**


#### 5.2.3. Algorithm Flow Chart

The flow chart of FaCO algorithm is shown in Figure 5. The mechanism of position updating is fused at each step within the maximum number of iterations until the optimal objective function value is output.

## 6. Experiment and Analysis

### 6.1. Parameter Setting

For the setting parameters in the algorithm, we based our decisions on the previous research on firefly and ant colony parameter setting (since we use the dataset Oliver30 as an example, the default number of cities is 30). We take the default conventional value for the parameters that have little effect on the result, and we determine its reference range and conduct experiments to determine the final parameter value for the parameters that have greater effects on the result. The parameters m, γ are set as follows. m is generally set to be between 1 and 1.5 times the number of cities to achieve better results, and γ is taken as a random number from 0 to 1 [32,33]. The parameters related to the ant colony are set as follows. The heuristic factor α, the pheromone volatility factor ρ and the expectation factor $\beta^{{}^{\prime }}$ are analyzed; when the city size is less than 100, p = [0.85n,1.20n] can obtain better results. The overall performance of the algorithm is better when α = [0.5, 0.9] in the case of n equal to 30. The best results are obtained when $\beta^{{}^{\prime }}$ = 2, 3, 4, 5. The best results are obtained when ρ = [0.4, 0.7] [34]. Therefore, we conducted the experiment with that result as the initial range.

In order to qualitatively evaluate our parameters [35], we use the control variables method to control the other parameters as constants so that α, $\beta^{{}^{\prime }}$, ρ, and γ vary in the given range, respectively, and obtain the objective function results as shown in Figure 6, Figure 7, Figure 8 and Figure 9. From the above figures, we can parameterize that the algorithm has the best search results when α = 0.5, $\beta^{{}^{\prime }}$ = 2, ρ = 0.69, and γ = 0.4.

### 6.2. Experiment Analysis

The improved FaCO algorithm incorporates the pheromone roulette selection of the ant colony algorithm into the brightness-dependent selection mechanism of fireflies, adds the selection of new population probabilistic merit on the basis of individual population meritocracy, and improves the merit-seeking efficiency of the algorithm on the basis of ensuring the merit-seeking rule of fireflies.

The FaCO algorithm proposed in this paper is analyzed experimentally and comparatively to verify the effectiveness of the algorithm (as shown in Table 2). It was shown experimentally that better results were achieved on the three data sets compared to the firefly algorithm itself as well as the particle swarm algorithm and the genetic algorithm. Moreover, compared with the firefly algorithm itself, the situation of falling into local optimum is avoided in the process of finding the optimum.

In this paper, we take the TSP public dataset Oliver30 as an example and obtain the objective function results as follows.

The results in Figure 10 and Figure 11 show that, in the same case, the firefly algorithm falls into the local optimum problem when it reaches about one hundred iterations, and the result remains flat at about 952. Although the FaCO algorithm, with the same number of iterations, has reached about 952 when it reaches about one hundred iterations, it does not fall into the local optimum solution but jumps out to continue the search until it reaches about 902. The FaCO algorithm is better than the firefly algorithm in terms of both the efficiency of the search and the solution to the local optimum and the search result; therefore, it is an effective and efficient improvement algorithm.

## 7. Conclusions

We use a multi-level graph dissection algorithm to coarsen the original complex graph structure into a simple graph structure TSP problem with a certain mapping strategy by converting the load-balancing problem of parallel computing into a graph network assignment problem. An innovative proposal is made of an improved FaCO algorithm with a weighted fusion of the iterative mechanism of the firefly algorithm and ant colony algorithm, which gets rid of the limitation of the single merit-seeking mechanism of the previous heuristic algorithm and combines the advantages of both for optimization improvement. TSP public data set Oliver30, eil51, etc. are used as experimental data, and the parameter range selection experiments and results confirm that the FaCO algorithm is significantly better than the firefly algorithm and particle swarm algorithm in terms of optimization results in this paper. The number of iterations of the FaCO algorithm is also less than that of the firefly algorithm when obtaining the same results. It maintains the global search capability of the heuristic algorithm and also reduces the spatial search breadth. Additionally, the mapping strategy of the graph dissection algorithm ensures the influence of the obtained search results on the initial complex problem.

## 8. Future Work

In this paper, we proposed the incorporation of fireflies in heuristic algorithms into ant colony algorithms and proposed weighted combination on the search rules to improve the search efficiency. The main focus is on the load-balancing problem after the multi-level graph dissection algorithm coarsens the complex graphs into simple graphs after task assignment without optimizing the performance of the graph dissection itself. In addition, the computation time of FaCO itself proposed in this paper increases as the computation scale increases. Therefore, it is also of great significance to pay more attention to the spatio-temporal dimension of the search path [36] and to optimize the computational performance of the algorithm itself and the load-balancing problem of parallel computation of the graph dissection process.

The following aspects can be optimized in the next step: (1) we can optimize and innovate the mapping strategy of the multilevel graph dissection algorithm to make it more suitable with the coarsened assignment results and improve the computational accuracy; (2) we can continue to study the parallelism mechanism of other heuristic algorithms so that they can better maintain the breadth of the search space while having a greater possibility of obtaining better results.

## Figures and Tables

**Figure 1 biomimetics-07-00168-f001:**
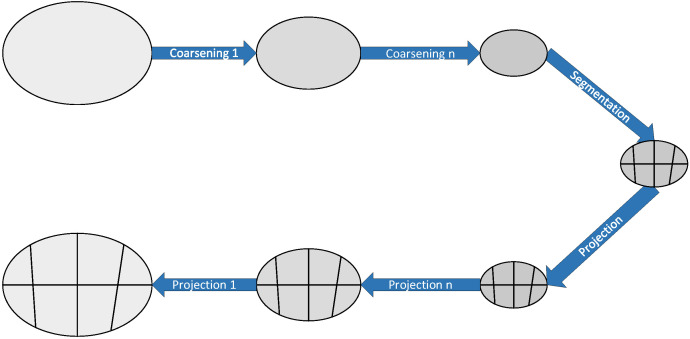
The coarsening process.

**Figure 2 biomimetics-07-00168-f002:**
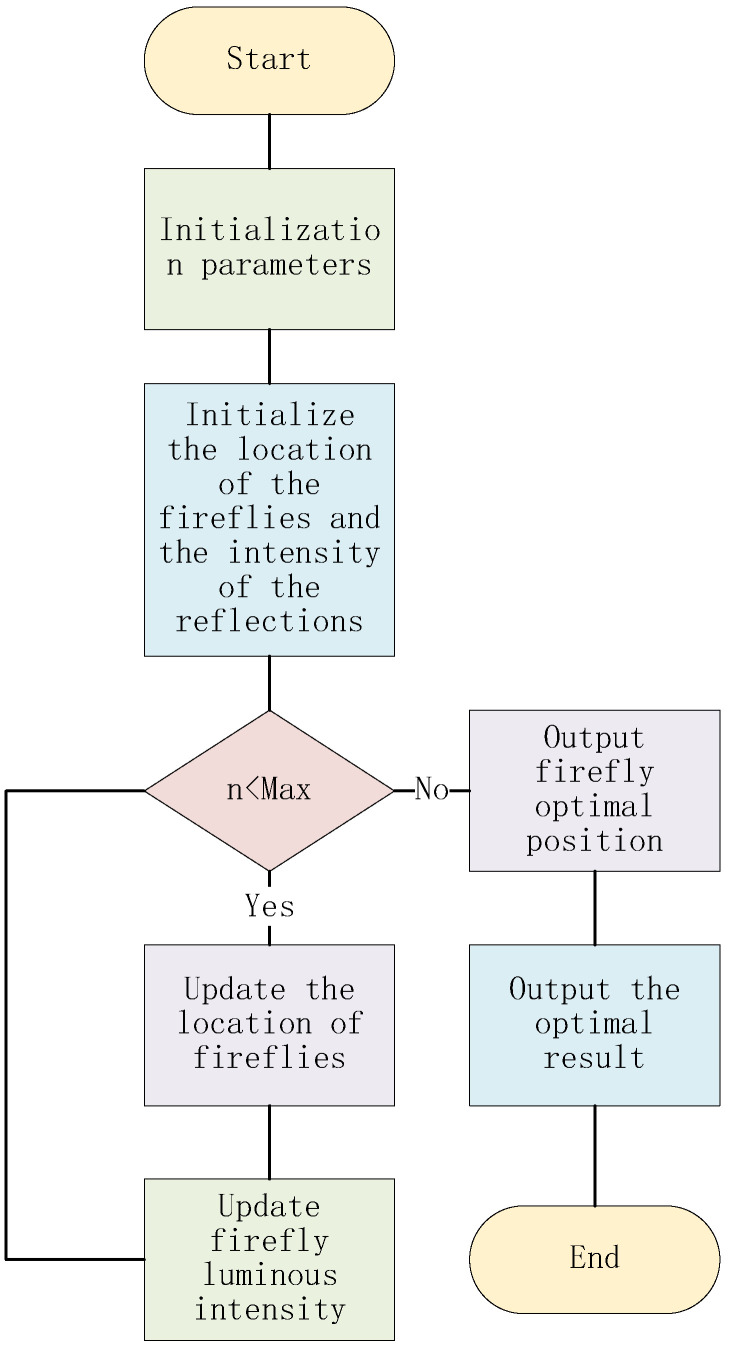
The flow chart of the firefly algorithm.

**Figure 3 biomimetics-07-00168-f003:**
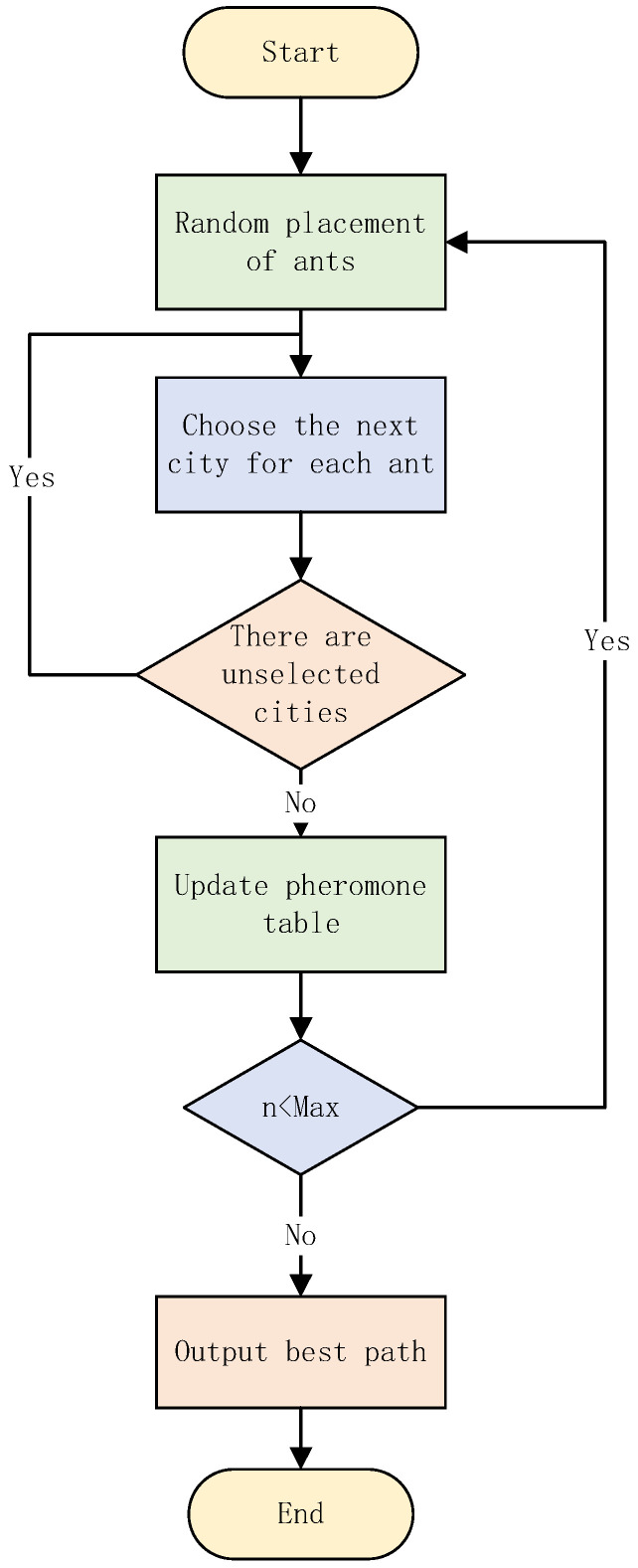
The flow chart of the ant colony optimization algorithm.

**Figure 4 biomimetics-07-00168-f004:**
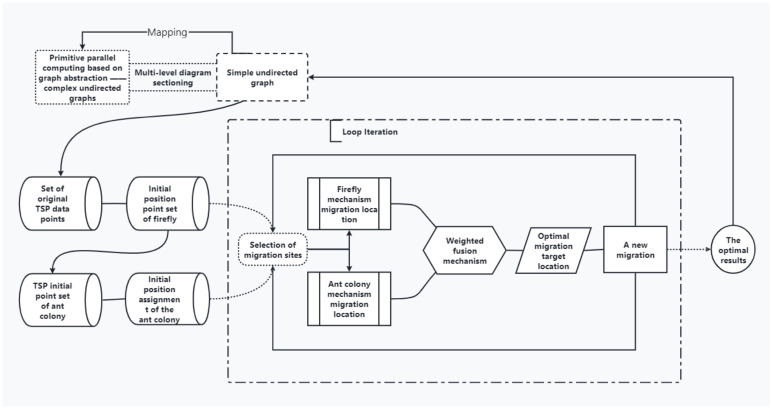
Logic diagram of the FaCO algorithm.

**Figure 5 biomimetics-07-00168-f005:**
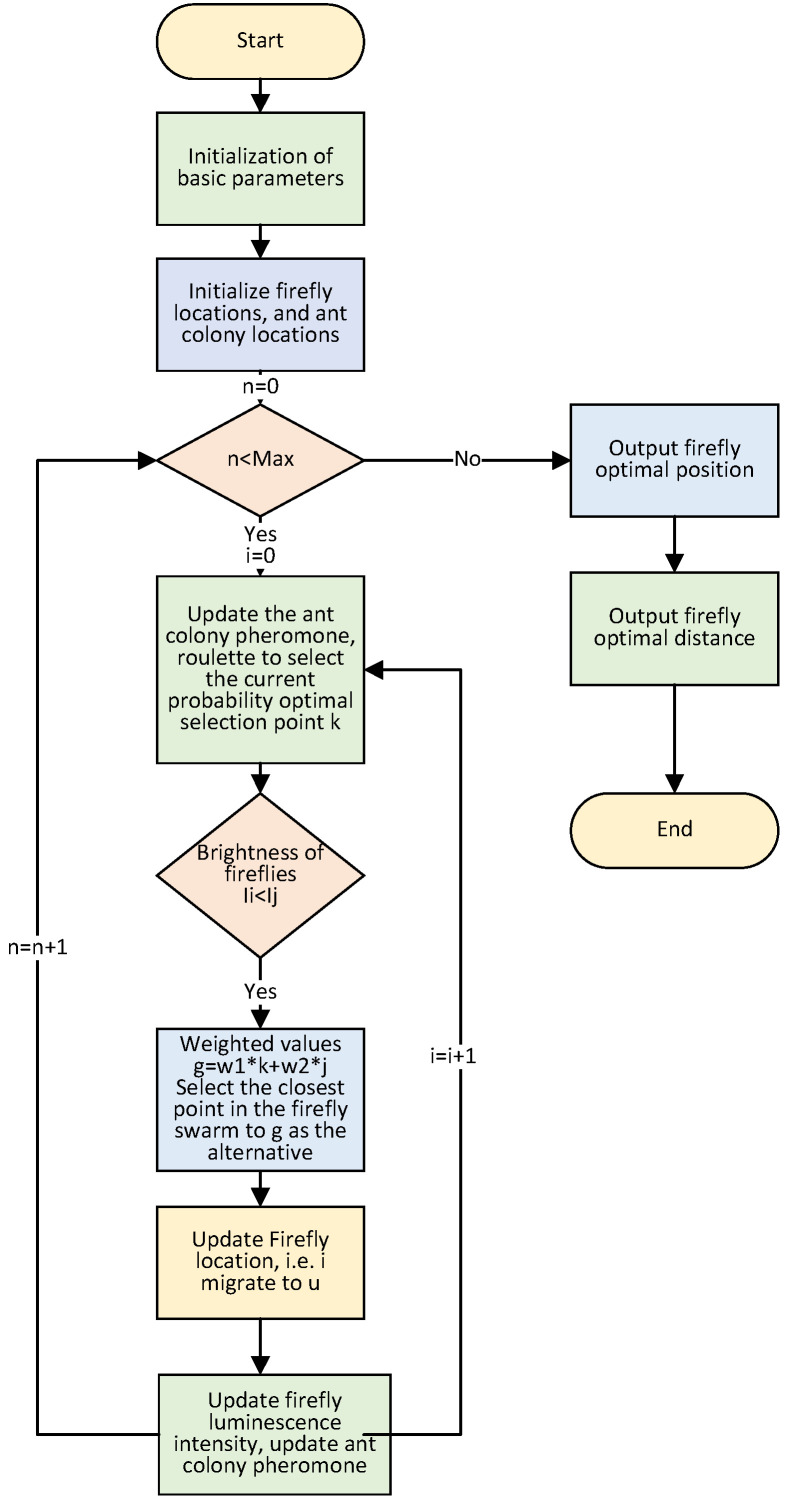
The flow chart of the FaCO.

**Figure 6 biomimetics-07-00168-f006:**
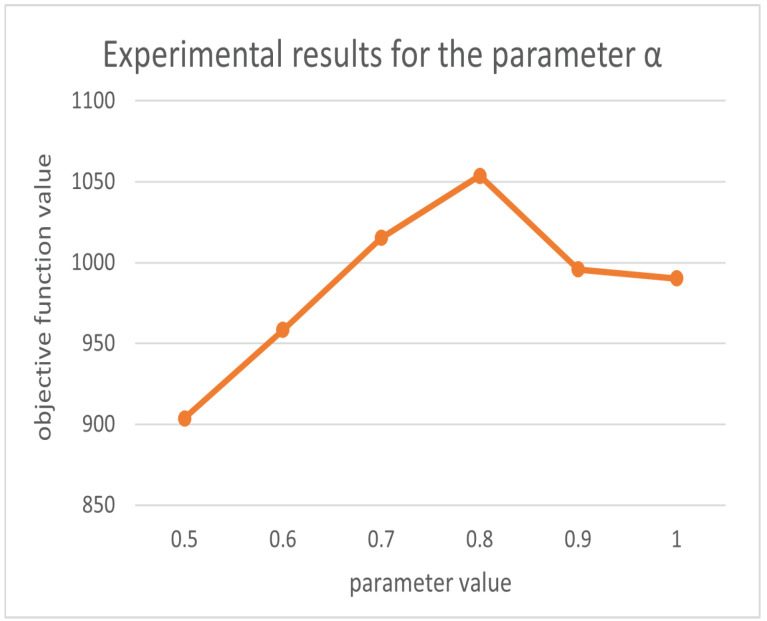
Experimental results for parameter α.

**Figure 7 biomimetics-07-00168-f007:**
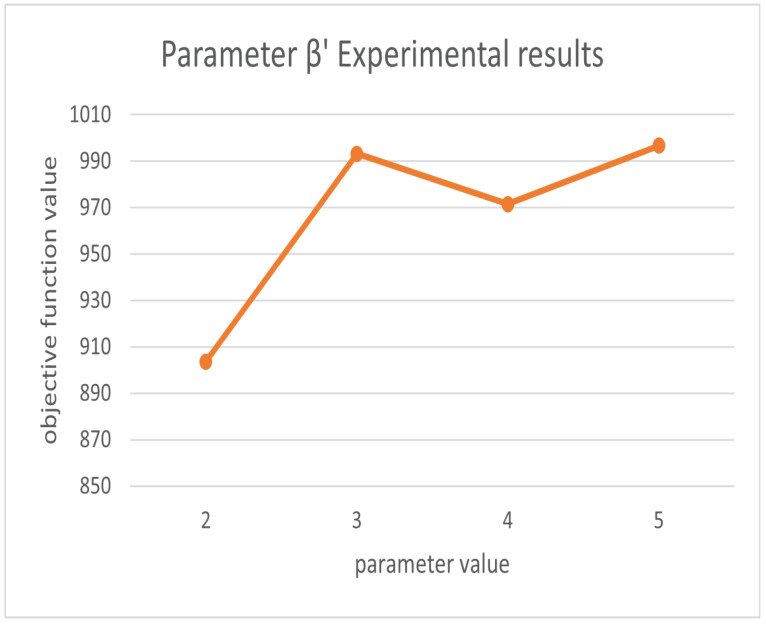
Experimental results for parameter $\beta^{{}^{\prime }}$.

**Figure 8 biomimetics-07-00168-f008:**
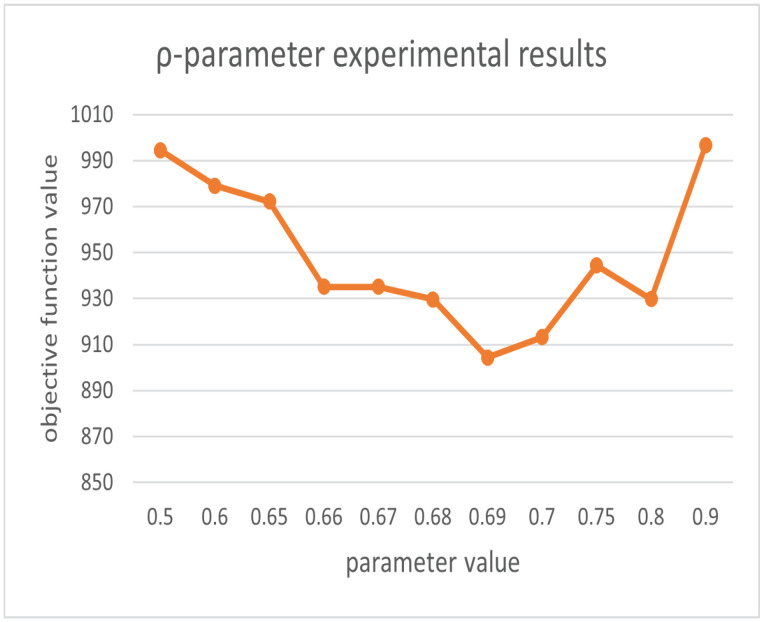
Experimental results for parameter ρ.

**Figure 9 biomimetics-07-00168-f009:**
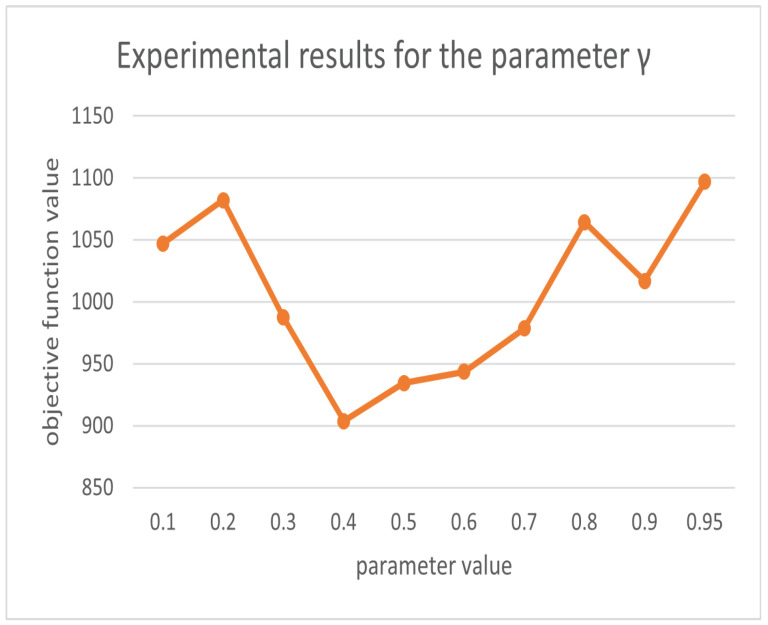
Experimental results for parameter γ.

**Figure 10 biomimetics-07-00168-f010:**
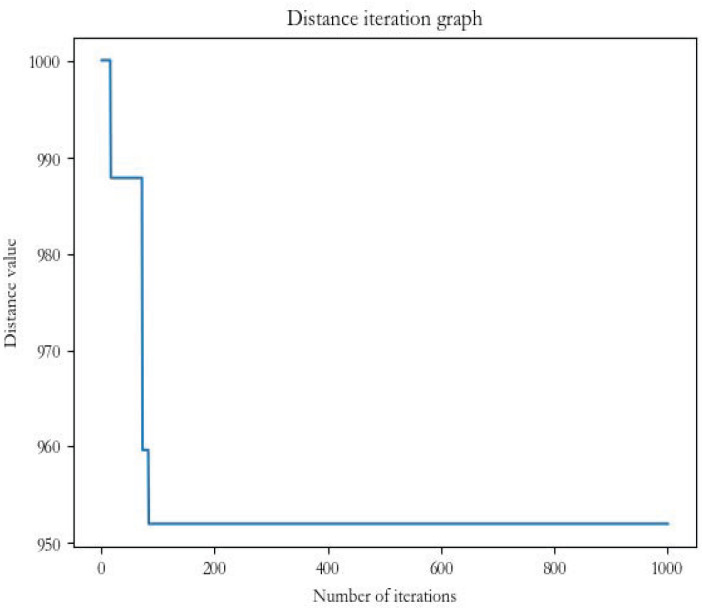
Firefly distance iteration chart.

**Figure 11 biomimetics-07-00168-f011:**
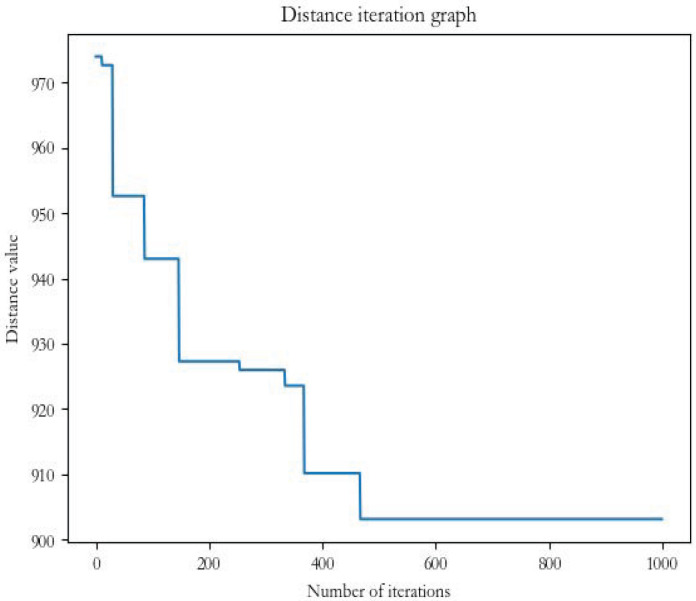
FaCO distance iteration chart.

**Table 1 biomimetics-07-00168-t001:** The initial parameters of the FaCO algorithm.

Name of Parameter	Meaning of Parameters	Remarks	Experimental Values
m	Firefly population size	1.5 times the number of cities	60
I${}_{0}$	Absolute brightness of fireflies	Objective function value correlation; the better the value of the objective function, the higher the brightness of their own	Determined by the initial position of the firefly
$\beta_{0}$	The maximum attractiveness of fireflies	The maximum attractiveness of fireflies when r=0	Generally set to 1
γ	Firefly light absorption coefficient		0.4
p	Colony population size	1.5 times the number of cities	60
Q	Colony pheromone constant	Usually takes the value of [10, 1000]	20
α	Colony pheromone factor	Range usually: [0.5, 1]	0.5
β′	Ant colony heuristic function factor	Range of values: [0, 5]	2
ρ	Ant colony pheromone volatility factor	Range of values usually: [0.4, 0.7]	0.69
Max	Maximum number of iterations		1000
r	Distance between two points		
$W_{1}$	Ant colony position update weight		0.35
$W_{2}$	Firefly position update weight		0.65

**Table 2 biomimetics-07-00168-t002:** Results of objective functions using the three datasets for the above-mentioned bioinspired algorithms.

TSP Arithmetic Example	Firefly Algorithm Results	FaCO Algorithm Results	Particle Swarm Algorithm Results	Genetic Algorithm
Burma14	44.129	30.8785	32.7	30.847
Oliver30	956.26	864.37	897.4	906.57
eil51	1436.788	807.175	823.6	842.1

## Data Availability

The code used during the running and testing of the algorithm is available in the file “FAACO_source” in the author’s GitHub repository https://github.com/DissertationResources/FAACO_source.git (accessed on 15 August 2022).

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
