# Peer review of "Load Balancing Based on Firefly and Ant Colony Optimization Algorithms for Parallel Computing"

_biomimetics, 2022, doi:10.3390/biomimetics7040168_

Round 1
Reviewer 1 Report
The problem under discussin in interesting and the proposed method is original. However, the paper needs improvements.
1- Add in the abstract and in the introduction the results and data used to solve this problem.
2- Add a last paragraph in the introduction to show the organization of this paper.
3- Section 2 displays the background and not the related work. Related works are not presented.
4- the flowchart of Ant (figure 3) is not clear, something is missing or wrong.
5- It is better to add a representation showing the combination of Ant and firefly (in addition to figure 4)
6- I suggest to add a new section to well explain the problem statement with some representations. There many notions parallel computing, load balencing, graph, bio-inspired. The reader could not follow the authors.
7- Table 2 shows the results of both algorithms, which results, do you mean the fitness function values? precise it please.
8- The datasets used should de defined.
9- What about setting parameters. The athors didn't show how the parameters are set? An experimental study should be performed for that.
10- The experimental study is very week, it contains one table and two figures with a very small paragraph. More experiments should be performed taking high dimentional data and starting with parameter setting.
11- The conclusion needs improvments. Future work are not mentioned.
Author Response
Dear reviewers, thank you very much for your review and suggestions on my article, and my response to you is uploaded as a file for review.

Reviewer 2 Report
In this paper, the authors have proposed a hybrid algorithm for parallel commuting by combining the firefly and ant colony optimization algorithms. The proposed work is interesting, but it is not clear what is the key novelty. It is a highly derivative result. It will be more suitable for conferences or technical reports, not as a full journal article.
Here are my general comments to the authors,
1) What is the key novelty of this work? It seems a hybrid approach of both algorithms has been applied to multiple fields like cloud computing, motion planning etc
2) Many acronyms have been used throughout the paper, but no mention of their full abbreviation, such as FAACO, TSP etc.
4) Please add some quantitative results in the abstract
5) In line 38, "at home and abroad," what does it mean? It is not standard practice in journal writing.
6) In line 40, again, "abroad". Please modify accordingly.
7) It is not clear why the authors have written sections 2.2 and 2.3. It seems they are unnecessary. Just paper citations would have been enough.
8) Why did the authors choose the results to compare only with the firefly algorithm, although there are other methods too, such as heuristic models etc.? Please compare the proposed algorithm results with other algorithms too.
9) The paper is poorly written, and it requires extensive editing. Please follow standard journal writing practices.
Author Response
Dear Reviewer, Thank you very much for your review and suggestions on my article. I have responded to each of your suggestions in the form of a document, and I hope that I can meet your expectations

Round 2
Reviewer 1 Report
Thank you for considering my comments. The article has been enhanced but there are some points that deserve attention.
- Each axis in the figures 6,7,8,9 should have a label (y axis is objective function value, and x is parameter value )
- The caption of Table 2 can be modified to "Results of Objective functions using three the datasets for the above mentioned bioinspired algorithms".
-the concept of parallel computing, the bioinspired algorithms, and graphs are not detailed. The explanation is very brief.
-the authors are confusing between literature review (where the related studies should be described) the background (where the main notions related to the problem should defined such that graphs, bioinspired algorithms, etc) and problem statement (where the problem under study should defined). Section 2 is not related work because only one reference is cited (one related work). It is rather a merge between background and problem statement.
- Related work section is required.
- the diagram presented in section 4.2 is very vague. It needs more involvement.
- Proofreading is needed
Author Response
Dear Reviewer, I am very honored to receive your second comment and thank you very much for your careful guidance of my paper. Each of your comments is very valuable to me, so I have made targeted revisions. My response is in the file I have attached.

Reviewer 2 Report
1. Please label the axes of figs 6-8
2. The paper still requires careful editing
3. Please modify the table 2 caption. It is very generic
Author Response

(The authors gave the same response as above.)
